# Phytopathological Threats Associated with Quinoa (*Chenopodium quinoa* Willd.) Cultivation and Seed Production in an Area of Central Italy

**DOI:** 10.3390/plants10091933

**Published:** 2021-09-16

**Authors:** Giovanni Beccari, Mara Quaglia, Francesco Tini, Euro Pannacci, Lorenzo Covarelli

**Affiliations:** Department of Agricultural, Food and Environmental Sciences, University of Perugia, Borgo XX Giugno, 74, 06121 Perugia, Italy; giovanni.beccari@unipg.it (G.B.); francesco.tini@collaboratori.unipg.it (F.T.); euro.pannacci@unipg.it (E.P.); lorenzo.covarelli@unipg.it (L.C.)

**Keywords:** downy mildew, *Fusarium*, FIESC, *Peronospora variabilis*, quinoa, seed

## Abstract

In 2017, in a new *Chenopodium quinoa* cultivation area (Central Italy), emergence failures of the Titicaca, Rio Bamba, and Real varieties, whose seeds were obtained the previous year (2016) in the same location, were observed. Moreover, leaf disease symptoms on the Regalona variety, whose seeds came from Chile, were detected. Visual and microscopic analyses showed the presence of browning/necrotic symptoms on the seeds of the three varieties whose emergence in the field had failed. In addition, their in vitro germination rates were strongly compromised. *Fusarium* spp. was isolated with high incidence from Titicaca, Rio Bamba, and Real seeds. Among the detected *Fusarium* species, in the phylogenetic analysis, the dominant one clustered in the sub-clade *Equiseti* of the *Fusarium incarnatum-equiseti* (FIESC) species complex. Instead, the pathogen associated with Regalona leaf symptoms was identified, by morphological and molecular features, as *Peronospora variabilis*, the causal agents of downy mildew. This is the first report of both *P. variabilis* and *F. equiseti* on *C. quinoa* in Italy. Species-specific primers also detected *P. variabilis* in Regalona seeds. These results underline the importance of pathogen monitoring in new quinoa distribution areas, as well as of healthy seed production and import for successful cultivation.

## 1. Introduction

Quinoa (*Chenopodium quinoa* Willd.) is an annual dicotyledonous seed-producing plant belonging to the *Amaranthaceae* family [1]. This species originates from South America, where it was first domesticated, presumably more than 7000 years ago, in the Andean region, near Lake Titicaca [2]. The same area acts also as a natural quinoa germplasm bank, since several quinoa varieties have here been differentiated and preserved over time by the indigenous populations [3]. Due to the high level of genetic diversity, the crop is highly resilient to agro-ecological extremes (soil, rainfall, temperature, and altitude) and it is tolerant to frost, drought, and soil salinity [2,4,5]. For this reason, quinoa can be grown on marginal lands unsuitable for other major crops, providing in these disadvantaged areas food of high nutritional value [6]. Indeed, quinoa seeds are gluten-free, with a low glycemic index, and contain an excellent balance of all nine essential amino acids, with high concentrations of histidine, lysine, and methionine; moreover, they are rich in fiber, lipids, carbohydrates, vitamins, minerals (including calcium, magnesium, and iron), and health-promoting compounds (flavonoids) [1,7,8]. Due to its agricultural and nutritional aspects, since the end of the 20th century, quinoa has gained international attention [9] to such an extent that the United Nations declared 2013 as the “International Year of Quinoa” [10] and in 2017 its genome was completely sequenced [6]. Therefore, to address the major market requests, a significant increase in the cropped area and production has been recorded during the last ten years [11]. Until 2018, quinoa was considered a major crop only in Bolivia (70,763 t, 111,605 ha), Peru (86,011 t, 64,660 ha), and Ecuador (2146 t, 2048 ha) [11,12]. However, quinoa is currently cultivated in all continents [13] because there have been numerous attempts at quinoa introduction from the area of origin to other countries, including the European ones. The number of countries growing the crop has increased from 8 in 1980, to 40 in 2010, to 75 in 2014 [2]. In 2018, quinoa was present for research and production in 123 countries, even if 74% of global exports are still supplied by Peru and Bolivia [14]. In Italy, quinoa was introduced in the early 2000s thanks to several research projects dealing with the adaptation of this species to the peninsula agricultural system [15,16,17]. Results indicated that, given its high resiliency and tolerance to abiotic stresses, in particular salinity and drought, quinoa could represent a good alternative to the traditional Mediterranean crops in light of the current climate change [18].

However, quinoa is susceptible to several biotic stresses that could strongly impair yield, both in the area of origin and in those where the crop has spread. Phytopathogenic fungi and oomycetes are among the main biotic stress factors affecting quinoa crops. In the Andean region, quinoa yield losses caused by several phytopathogens have been well documented [19,20]. In this geographical area, downy mildew is undoubtedly the most diffused and well-known disease and the epidemics, caused by the heterothallic oomycete *Peronospora variabilis* (Gaüm), greatly reduce seed yield [21]. In addition to downy mildew, other quinoa diseases were also detected in the Andean area, such as leaf spots caused by *Ascochyta hyalospora* (Cooke and Ell.) and *Ascochyta chenopodii* Rostr., black stem caused by *Ascochyta caulina* [P. Karst. (v.d. Aa and v. Kest.)], leaf spot caused by *Cercospora* spp., brown stalk rot by *Phoma exigua* var. *foveata* [(Foister) Boerema], and damping-off of root/seedlings by *Rhizoctonia* spp., *Fusarium* spp., and *Pythium* spp. [12,19,20,22]. The incidence and the severity of the various diseases caused by fungi and oomycetes depend on different factors, such as environmental conditions, the cultivated variety, and the phenological stage of the crop during which the infection occurs [20]. With the rapid expansion of quinoa growing areas, the problems of the negative impact of some of the above-reported phytopathogens have arisen also in the new countries [23]. Therefore, with the extension of quinoa cultivation, traditional and/or new pathogens may threaten the spread of this species. Consequently, understanding the phytosanitary problems related to this crop in new cultivation areas is a key step to efficiently counteract and manage them.

For this reason, in a new quinoa cultivation area (Umbria, Central Italy), after observing in 2017 emergence failures of seedlings belonging to the Titicaca, Rio Bamba, and Real varieties, as well as leaf symptoms on the Regalona variety, a series of phytopathological surveys, both on seed lots and on leaves, were carried out. The results of this survey are presented in this study and can be useful to better manage quinoa diseases in Italy, both for healthy seed production, as well as for its successful cultivation. 

## 2. Results

### 2.1. Examination of Sampled Material

#### 2.1.1. Seed Material

A combination of visual and stereomicroscopic observations of seed material allowed the detection of the presence of browning/necrotic symptoms on the seeds of the Titicaca, Rio Bamba, Real, and Regalona varieties, with the seeds of the first three varieties showing symptom incidences of 93%, 77%, and 70%, respectively, which were significantly higher (*p* ≤ 0.05) than those recorded in the seeds of the Regalona variety (17%) (Figure 1). In the first three varieties, seed symptoms were uniformly distributed almost on the whole seed surface and were more severe compared to those recorded on the Regalona ones, showing only small browned/necrotic areas (Figure 1).

#### 2.1.2. Plant Material

Visual examination of Regalona plant material showed the presence of foliar chlorosis/yellowing symptoms (Figure 2a,b). In detail, a combination of visual and stereomicroscopic observation showed that the upper side of symptomatic leaves was characterized by yellow/chlorotic spots with corresponding black/grey efflorescence on the adaxial surface (Figure 2b–d). Microscopic observation of this efflorescence showed the presence of colorless zoosporangiophores subdichotomously branched, slightly curved, with obtuse tips, each carrying a single pale brown ellipsoidal zoosporangium with a short pedicel (Figure 2e). These features led us to hypothesize that the pathogen belonged to the *Peronospora* genus inside the *Oomycetes* class of the *Pseudofungi* subphylum of the *Chromista* kingdom [24], thus molecular investigations were carried out to complete the identification (Section 2.5).

### 2.2. Seed Germination Rate

Within each seed category (symptomatic and asymptomatic), significant differences (*p* ≤ 0.05) in germination rate among the four varieties were detected (Figure 3). In detail, considering asymptomatic seeds, the Regalona variety showed the highest (*p* ≤ 0.05) germination rate (100%), followed by Real (70%), Rio Bamba (40%), and Titicaca (36.6%) (Figure 3). Only the germination rate of asymptomatic seeds of the Real variety was not significantly different (*p* ≥ 0.05) from that of Regalona (Figure 3). Considering symptomatic seeds, no germination was recorded in the Titicaca, Rio Bamba, and Real varieties, while symptomatic seeds of the Regalona variety showed a germination rate of 73.3% (Figure 3).

Within each variety, significant differences (*p* ≤ 0.05) in germination rate were also recorded among the two seed categories (Figure 3). In all varieties, asymptomatic seeds showed significantly higher (*p* ≤ 0.05) germination rate levels than those of symptomatic ones.

### 2.3. Seed Mycological Analysis

The combination of visual, stereo-, and light-microscope observations directly on isolation plates allowed us to identify the mycoflora composition of each variety and seed category. In general, *Alternaria* and *Fusarium* were the most representative fungal genera, being detected in each seed category of the Titicaca, Rio Bamba, and Real varieties (Figure 4). With a lower incidence, the genera *Aspergillus* and *Penicillium* were also detected in the seeds of these three varieties, with *Aspergillus* detected only on asymptomatic seeds and never on the symptomatic seeds of these three lots, in which, however, healthy seeds were absent. On the contrary, apart from the negligible presence of *Alternaria* and *Penicillium* genera, a high incidence of healthy seeds was detected in each seed category of the Regalona variety, where *Fusarium* and *Aspergillus* were not detected (Figure 4).

Significant differences (*p* ≤ 0.05) between the incidences of different fungal genera within each “variety–seed category” combination were recorded (Figure 4). Indeed, in the seeds of Titicaca, Rio Bamba, and Real varieties, colonies belonging to the genera *Alternaria* and *Fusarium* always showed an incidence higher than those of the other genera (Figure 4). These differences were “mitigated” in the asymptomatic seed category but, conversely, they were stronger in the symptomatic seed category, where, in addition, *Fusarium* incidence (87%, 70%, and 97% for Titicaca, Rio Bamba, and Real varieties, respectively) was always significantly higher (*p* ≤ 0.05) than those of *Alternaria* (40%, 27%, and 40% for Titicaca, Rio Bamba, and Real varieties, respectively) (Figure 4). *Fusarium* was absent in all the two categories of Regalona seeds where, differently from the other varieties, the incidences of healthy seeds were always significantly higher in each seed category (90% and 73% for asymptomatic and symptomatic seeds, respectively) (Figure 4).

Significant differences (*p* ≤ 0.05) among the incidences of different fungal genera within each “variety–fungal genera” combination were also detected. In particular, *Fusarium* incidence in symptomatic seeds of Titicaca, Rio Bamba, and Real varieties (87%, 70%, and 97%, respectively, as above reported) was significantly higher (*p* ≤ 0.05) than that recorded in asymptomatic seeds (33%, 37%, and 27%, respectively). No significant differences (*p* ≥ 0.05) of *Aspergillus* and *Penicillium* incidences were detected among the different seed categories of Titicaca, Rio Bamba, and Real. Similarly, the incidence of healthy seeds was not significantly different (*p* ≤ 0.05) among the two different seed categories in the Regalona variety (Figure 4).

Finally, significant differences (*p* ≤ 0.05) between the incidences of the different fungal genera within each “fungal genera–seed category” combination were also recorded. The incidences of the *Alternaria* genus in both seed categories of Titicaca, Rio Bamba, and Real varieties were significantly (*p* ≤ 0.05) higher than those detected in the same two seed categories of Regalona varieties. In addition, *Fusarium* incidence in the symptomatic seed of the Rio Bamba variety was significantly lower (*p* ≤ 0.05) than those recorded in the same category of Titicaca and Real varieties.

### 2.4. Molecular Identification of Fusarium *spp.* Associated to Quinoa Seeds

The BLAST analysis indicated that the *Fusarium* community isolated from quinoa seed lots of the Titicaca, Rio Bamba, and Real varieties, multiplied in 2016 at Papiano (42°57′ N, 12°22′ E, 165 m a.s.l., Perugia, Umbria, Central Italy), was composed of a total of four species: *Fusarium incarnatum-equiseti* species complex (FIESC), *Fusarium avenaceum* [(Fr.) Sacc.], *Fusarium culmorum* [(Wm.G.Sm.) Sacc.], and *Fusarium sporotrichioides* (Sherb.) (Figure 5). Interestingly, FIESC was the most frequent species of the *Fusarium* complex in each variety–seed combination category (*p* ≤ 0.05). Indeed, the other three species were detected with an incidence that ranged from 1% to 4% (Figure 5).

Therefore, FIESC incidence was investigated more in-depth. In detail, within each seed category (symptomatic, asymptomatic), no significant differences (*p* ≥ 0.05) in FIESC incidence between the three varieties (Titicaca, Rio Bamba, and Real) affected by FIESC were detected (Figure 6), while, within each variety, significant differences (*p* ≤ 0.05) in FIESC incidence between the two seed categories were observed. In particular, in Titicaca, Rio Bamba, and Real varieties, symptomatic seeds showed FIESC incidences (80, 66, and 86%, respectively) significantly higher (*p* ≤ 0.05) than those recorded in asymptomatic seeds (30, 33, and 26%, respectively) (Figure 6).

To assess the impact of FIESC incidence on quinoa seeds’ germination rate, the correlation between these two parameters was evaluated (Figure 7). Taking the Titicaca, Rio Bamba, and Real varieties (the only ones in which FIESC were detected) and each seed category together, seed germination was negatively and significantly related to FIESC incidence (R^2^ = 0.62; *p* = 0.004). Considering individual varieties, in Rio Bamba and Titicaca, the negative association between germination rate and FIESC incidence was statistically significant (*p* = 0.02 and 0.03, respectively). Finally, considering each seed category individually, the strongest negative association was observed for symptomatic seeds (*p* = 6 × 10^−11^), since the absence of germination and the highest FIESC incidence was detected in the seeds of this category for the Titicaca, Rio Bamba and Real varieties.

Due to the negative impact of FIESC on the germination rate of the quinoa seeds belonging to the varieties Titicaca, Rio Bamba, and Real, the FIESC representative isolate, denominated FIESC-PG-Q1, was further molecularly characterized (Figure 8). Amplification of the *translation elongation factor 1α* (*tef1α*) of FIESC-PG-Q1 isolate produced a fragment of 700 bp. According to Reference [25], in the phylogram constructed on the concatenated sequences of *tef1α* region of the validated phylogenetic species of FIESC, two major clades emerged: *Equiseti* clade, including the phylogenetically validated species from FIESC 1-a to FIESC 14-a and 30-a (Appendix A) and *Incarnatum* clade, including the phylogenetic validated species from FIESC 15-b to FIESC 29-a. Isolate FIESC-PG-Q1 (MZ191105 GenBank accession, Appendix A) clustered in a sub-clade of the *Equiseti* clade together with the phylogenetically validated species FIESC 5 (a, b, c, d, and f) (Figure 8).

### 2.5. Molecular Identification of Pathogen Infecting Quinoa Leaves and Its Detection Seeds

Amplification of the *Internal Transcribed Spacer* (*ITS*) region of PV-PG-Q1 isolate produced a fragment of about 280 bp. Through preliminary BLAST analysis, this isolate was attributed both to *Peronospora variabilis* and to *Peronospora farinosa,* thus, a phylogenetic analysis was carried out to uniquely attribute it to a certain species. In the phylogram built on the concatenated sequences of the *ITS* region of validated *Peronospora* species from *C. quinoa* and other plant species belonging to the *Amaranthaceae* family (Appendix A), three major clades (A, B and C) emerged (Figure 9). Clade A contained isolates of *P. variabilis* from *Chenopodium album* L. and *C. quinoa*; clade B contained isolates of *Peronospora chenopodii* Schltdl. from *Chenopodium hybridum* L., *Peronospora boni-henrici* Gäum. from *Chenopodium bonus-henricus* L., *Peronospora effusa* (Greville) Cesati from *Spinacia oleracea* L., and *Peronospora farinosa* from *Atriplex* spp.; clade C contained isolates of *Peronospora chenopodii-polyspermi* Gäum. from *Chenopodium polyspermum* L. (Figure 9). Isolate PV-PG-Q1 (MZ191106 GenBank accession, Appendix A) clustered together with *P. variabilis* isolates from *C. album* and *C. quinoa* (Figure 9).

Finally, the PCR protocol used with PV6F and PV6R primers also showed the presence of the same foliar pathogen in the seeds of the Regalona variety (presence of specific amplification product at 278 bp) belonging to the same batch used to set up the experimental field (Appendix A).

## 3. Discussion

The rapid expansion of quinoa from its area of origin into new cultivation areas could be accompanied by the crop exposition to the damaging effects of known and/or new pathogens [23]. In this context, monitoring quinoa in new growing areas is a determining factor to identify the phytosanitary threats of this crop and to plan efficient management strategies. For this reason, this work shows the results of a series of phytopathological surveys carried out after observing, in a new cultivation site (Umbria, Central Italy): (a) severe crop emergence failures in a number of quinoa varieties and (b) severe leaf disease symptoms on one of the surveyed varieties.

Our investigations, carried out on the same seed batches used to set up the field trial in which emergence failures were observed, allowed us to detect the presence of browning/necrotic symptoms on seeds, with the highest incidence and severity detected on the seeds of the three varieties (Titicaca, Rio Bamba, and Real) showing emergence failures. The germination rate of Titicaca, Rio Bamba, and Real seeds was strongly compromised if compared with that of Regalona variety, not only in symptomatic seeds but also in the asymptomatic ones. In addition, the mycobiota composition associated with the seeds showed substantial differences between Titicaca/Rio Bamba/Real varieties and the Regalona variety, with the latter almost free from fungal pathogens except for low levels of *Alternaria* spp. and *Penicillium* spp. Conversely, the fungal community isolated from Titicaca, Rio Bamba, and Real seeds showed a high incidence of microorganisms belonging to the genera *Fusarium* and *Alternaria*, with the first ones particularly abundant in symptomatic seeds. This difference in composition/incidence of the mycobiota associated with quinoa seeds could be attributable to the different areas in which the seeds were obtained. Indeed, as mentioned above, the seeds of Titicaca, Rio Bamba, and Real varieties were all obtained in the new Italian cultivation site (Umbria, Central Italy) in the year (2016) preceding the one (2017) in which emergence failures were observed, while the *Fusarium*-free Regalona seeds originated from Chile. Thus, the seed production area affected the composition of the fungal community associated with quinoa seeds.

The *Fusarium* community associated with seeds of Titicaca, Rio Bamba, and Real varieties was dominated by FIESC, which was particularly present in the symptomatic lots. In vitro seed germination rates, especially those of asymptomatic and symptomatic seeds, were negatively correlated to FIESC incidence, suggesting that this *Fusarium* complex could have potentially been implicated in the emergence failure observed in the field. In fact, the negative effect of *Fusarium* species on seed germination has been well documented [29,30].

However, the presence of other fungal genera and other *Fusarium* species detected during the present work makes it difficult to define with certainty the amount of seed germination and seedling emergence losses caused by FIESC and observed in the field. In addition, fungal pathogens could be only one of the factors that influence the germination and emergence of quinoa seedlings. Indeed, in addition to biotic factors (pathogens, but also pests), a complex of abiotic factors and agronomic conditions, such as soil condition, soil preparation before sowing, sowing depth, environmental conditions, low soil moisture content, and crusty topsoil [12,31,32], can also strongly reduce quinoa seed germination and plant emergence [33]. Further investigations could be useful to assess the impact of FIESC on seed germination and on quinoa emergence.

FIESC members are generally associated with diseases of several crops, particularly cereals [25,34,35]. They are considered “sporadic” and “weak” causal agents of *Fusarium* head blight (FHB) of wheat and barley. For example, in the same Italian cultivation area of the quinoa experiment described in this paper, FIESC has been often detected as a “minor” component of the FHB complex of wheat and barley [36,37,38]. In addition, leaf spots caused by FIESC have been observed in different Italian areas on leafy vegetable hosts (i.e., lettuce, rocket, spinach, etc.) [35]. FIESC members have also the ability to biosynthesize mycotoxins, including type A and type B trichothecenes, zearalenone, beauvericin, fusaric acid, and moniliformin [39,40], representing a risk for human and animal health.

FIESC is a phylogenetically species-rich complex that includes over 30 cryptic phylogenetic species, making identification based on phenotypic characteristics problematic [41]. The molecular analysis conducted in this study was able to ascribe the representative isolate FIESC-PG-Q1 in a sub-clade within the *Equiseti* clade together with the phylogenetic validated species FIESC 5. At least two distinct morphotypes have been reported in the literature for *Equiseti* clade species: morphotype I and II, with short or long apical cells, respectively [42]. Morphotype II is predominant in southern Europe [43] and has been associated to FIESC 5 [25]. Although FIESC 5 has also been previously detected by Reference [44] in Italian soil, to the best of our knowledge, this is the first report of FIESC 5 on *C. quinoa* seeds in Italy. *F. equiseti* has been reported in *C. quinoa* also in Brazil [45].

FIESC members are usually seed-borne pathogens [46] and *C. quinoa* seed production in Italy could be threatened by them. In fact, similarly to the main FHB agents, FIESC accumulation in cereal grains occurred following field infection at the head level (during the anthesis stage) and it could have been then exacerbated by inappropriate storage conditions. The presence of FIESC in *C. quinoa* seeds may follow a similar pathway; however, specific studies should be performed to assess the infection process of FIESC in quinoa inflorescence and/or seeds. In addition, the ability of this pathogen to produce mycotoxins reveals a possible threat to the safety of quinoa seeds destined for human consumption.

Several reports on *Fusarium* spp. isolation from diseased quinoa roots and young seedlings are available. *Fusarium* spp. were reported as one of the quinoa root rot and damping-off casual agents, often in association with *Rhizoctonia solani* (Cooke) Wint., *Pythium* spp., *Alternaria* spp., and *Acremonium* spp. [19,33,47,48]. Focusing on the genus *Fusarium*, damping-off and root rot diseases of quinoa are currently reported to be associated with *Fusarium solani* [(Mart.) Sacc.] and *Fusarium oxysporum* (von Schlechtendal) in Egypt [47,48] and *Fusarium avenaceum* in the Czech Republic [33]. The results of the present study contribute to expanding the knowledge of the *Fusarium* species that could be associated with *C. quinoa* in new cultivation areas. Microorganisms belonging to this genus could be common soil-inhabiting fungi that colonize the roots [35]. However, our results show that, in addition to soil-borne diseases, *Fusarium* spp. could also cause quinoa seed-borne diseases, as they were isolated from quinoa seeds before sowing.

The seeds of the Regalona variety analyzed during this study were completely free of *Fusarium* spp. However, the developed Regalona plants, which did not have any emergence problems, showed severe leaf symptoms (chlorosis and necrosis) and signs (sporangiophores) attributable to a downy mildew infection. The morphological features of the representative isolate PV-PG-Q1 obtained from symptomatic Regalona leaves matched with those described by References [49,50] for the species *Peronospora variabilis*. The identification of isolate PV-PG-Q1 as *P. variabilis* was also confirmed by molecular analysis.

As reported by several authors [20,22,51,52], *P. variabilis* has been specifically detected in *C. quinoa* in Bolivia, Chile, Colombia, Ecuador, Peru, USA, India, Canada, Turkey, Denmark, Korea, and Egypt, but not in Italy. The same pathogen was also reported on *C. album* in several countries (Peru, Argentina, Romania, Germany, Latvia, the Netherlands, Ireland, Korea, China, Turkey, Denmark) [21,22,50,53,54], including Italy [49]. Thus, to the best of our knowledge, this is the first report of *P. variabilis* on *C. quinoa* in Italy. Downy mildew is reported as one of the main causes of quinoa yield reduction all over the world; the losses depend on various parameters, such as the plant’s phenological stages at the infection time, presence of favorable weather conditions for the pathogen, and the level of cultivar susceptibility [55]. On a susceptible cultivar, in favorable weather conditions, a *P. varaibilis* attack during the first phenological stages can lead to total yield loss. As, unlike *Fusarium* spp., *P. variabilis* does not produce mycotoxins and does not affect the healthiness of the food product, the loss it causes is mainly quantitative.

Since the field experiment analyzed in this study had been sown with Regalona variety seeds, the presence of the pathogen in the seeds was hypothesized and confirmed by PCR assays carried out with species-specific primers (PV6F and PV6R), confirming *P. variabilis* as a seed-borne oomycete pathogen [20,22]. Indeed, while during *C. quinoa* growth in the field, *P. variabilis* are easily transmitted by low-distance zoospore dissemination by wind and rain, at long distances and between successive crop cycles the pathogen spreads by oospores [20]. However, given that *P. variabilis* is heterothallic, the presence of the two mating types P1 and P2 is required for sexual reproduction and, thus, oospores formation [20]. Previous research showed that oospores of *P. variabilis* were present in the seed pericarp [55]. Another study revealed that they were mainly localized in the perianth and seed coat (>85%), while only a very small percentage (<5%) were detected in the embryo and perisperm [56]. From the seeds, *P. variabilis* can move inside *C. quinoa* tissues, causing a systemic host colonization. A few days after seed germination, oospores were detected in the cortex of hypocotyls and the mesophyll of cotyledons [22,57]. The efficacy of infection transmission from the seed to seedling is favored by the high relative humidity at the sowing time, as well as by large oospore density [58]. This suggests that, in addition to the use of tolerant varieties, also avoiding the excess of water in the field and adjusting the space between rows, making the area less dense, are cultural practices that could contribute to reducing the risk of *P. variabilis* proliferation [20].

Thus, our investigation showed FIESC and *P. variabilis* are also important threats to quinoa cultivation in Italy and underlines the importance of healthy seed production, import, and use for successful quinoa cultivation. As already practiced for other crops (i.e., sunflower, cereals, etc.), the import and commercialization on a national and global scale of seed materials dressed with fungicides or, for organic farming, with bio-fungicides or heat-treated, would be desirable to avoid the spread of FIESC and *P. variabilis,* as well as of other seed-borne pathogens [20].

However, given the current lack of registered fungicides on quinoa in Italy, the management of these two pathogens (as well as of other pathogens) can negatively affect quinoa cultivation in both organic and integrated disease management approaches. Moreover, even if, to the best of our knowledge, no pathogenicity tests have ever been carried out to verify the infectivity of *P. variabilis* isolates from *C. album* on quinoa, the elimination of *C. album* weeds on the quinoa field would be advisable.

In fact, *C. album* is frequently infected by downy mildew throughout Europe because it is conspecific with the *P. variabilis* from *C. quinoa*; therefore, it is likely to be a reservoir for the pathogen and an alternative host [22]. Other *Chenopodium* species were reported to host the pathogen but these studies need further investigation. Cross-infection of *P. variabilis* on *C. album* and *C. quinoa* is the only instance that has been reported to date [22]. Mechanical and manual weed control is the most suitable method to control *C. album,* both in organic farming and in integrated pest management, because there are no selective herbicides registered for quinoa and other minor crops in Europe [17,59,60].

With the rapid expansion of quinoa growing areas, the negative impact of some other pathogens can arise in quinoa cultivation in Italy as well. For example, in addition to downy mildew, in the new cultivation regions, other fungal diseases such as gray mold caused by *Botrytis cinerea* (Pers.) [23], black stem caused by *Ascochyta caulina* [61], leaf spot caused by *Heterosporicola beijingense* sp. nov. [62], and anthracnose caused by *Colletotrichum nigrum* (Ellis and Halst) and *Colletotrichum truncatum* (Schwein) [22] were observed in the UK (gray mold), China (black stem and leaf spot), and USA (anthracnose), showing that diseases may easily occur when quinoa is introduced into nontraditional cultivation areas.

In conclusion, the introduction of a new plant species like *C. quinoa* in a new cultivation area requires the constant monitoring of its pathogens, with particular attention to those coming from other areas through infected seeds and those that arise as new threats in the new geographical distribution zone. In detail, these results highlight a particular need for the development of rapid, sensitive, and reliable methods to screen for quinoa seeds and plant pathogens. As suggested by Reference [22], this could be useful for the early detection of casual agents before diseases become too developed, but also to ensure certified pathogen-free quinoa seeds, which is an important requirement to be achieved at the global level.

## 4. Materials and Methods

### 4.1. Field Observations, Samples Collection, and Examination of Sampled Materials

In April 2017, in a quinoa field trial located at Papiano (42°57’ N, 12°22’ E, 165 m a.s.l.), near Perugia (Umbria, Central Italy; Figure 10), emergence failures of the Titicaca, Rio Bamba, and Real varieties were observed. Seed multiplication of these varieties had been performed during the previous year (2016) in the same location. Conversely, an optimal emergence was recorded for the seeds belonging to the Regalona variety, imported from Chile. However, on Regalona plants, foliar chlorotic symptoms were detected in the same field trial in June 2017, stimulating a series of phytopathological investigations on seed lots of the four varieties and on the leaves of the Regalona variety.

Thus, representative samples (30 g each corresponding to about 15,000 seeds on average) of the same four seed batches used to set up the field trial and a total of 30 plants of the Regalona variety, randomly chosen during the field trial, were collected. All samples (seeds and plants) were subject to a combination of visual and stereomicroscopic (SZX9, Olympus, Tokyo, Japan) observations to detect symptoms and, if present, signs of possible pathogens.

Concerning the four seed batches, during visual and stereomicroscopic observations, the incidence (%) of symptomatic seeds was assessed on a subsample of 30 randomly chosen seeds divided into three replicates (10 seeds per replicate).

For the Regalona samples, the formation of pathogen signs was promoted by placing a total of 10 symptomatic leaves, randomly collected from sampled plants, in humid chambers obtained by using 40 mL of sterile 1% water–agar (Biolife Italiana, Milan, Italy) in Petri dishes (150 mm diameter; Nuova Aptaca, Canelli, Italy). After 24 h of incubation under natural lighting, chlorotic leaves in humid chambers were observed as previously described. Moreover, pathogen signs were observed in more detail and photographed by a light microscope (Axiophot, Zeiss, Oberkochen, Germany).

### 4.2. Seed Germination Test

A total of 10 g (about 5000 seeds) of the above-reported seed batches of the Titicaca, Rio Bamba, Real, and Regalona varieties were randomly collected and used to assess the germination rate (%). In detail, for each batch, a total of 30 symptomatic and 30 asymptomatic seeds were randomly selected and divided into 3 replicates of 10 seeds each that were placed into 3 Petri dishes (90 mm diameter; Nuova Aptaca) onto two layers of sterile filter paper (Whatman N. 1, Maidstone, UK), previously added to 10 mL of sterile deionized water. Before assessing the incidence of germinated seed, Petri dishes were sealed with parafilm (Bemis Amcor, Oshkosh, WI, USA), to avoid water evaporation, and incubated for 6 days at 22 °C, in the dark.

### 4.3. Seed Mycological Analysis

For each of the four varieties (Titicaca, Rio Bamba, Real, and Regalona) a total of 30 symptomatic and 30 asymptomatic seeds were taken from a randomly collected seed sample of 10 g and used to assess the mycobiota associated with each seed batch. In detail, seed mycological analysis was performed as previously indicated by Reference [36], adapting this method to quinoa seeds. Briefly, seeds were externally disinfected for 2 min using a water–ethanol (95%, Sigma Aldrich, Saint Louis, MO, USA)-sodium hypochlorite (7%, Carlo Erba Reagents, Milan, Italy) solution (82:10:8% vol.) and rinsed with deionized sterile water for 1 min. Thirty seeds were placed onto potato dextrose agar (PDA; Biolife Italiana) pH 5.7, supplemented with streptomycin sulfate (0.16 g/L, Sigma Aldrich) into 3 Petri dishes (90 mm diameter, Nuova Aptaca) (replicates) containing 10 seeds each, for a total of three replicated plates per variety. The dishes were incubated at 22 °C in the dark.

After 6 days of incubation, a combination of visual and stereomicroscopic (SZX9, Olympus) observations were carried out on each seed to assess the presence of the different developed fungal genera. Light-microscopic (Axiophot, Zeiss) observations of fungal structures characterizing the developed colonies were also performed.

### 4.4. Molecular Identification of Fusarium *spp.* Associated to Quinoa Seeds

Fungal isolates obtained from seed samples were transferred in pure cultures into new plates containing PDA and placed at 22 °C, in the dark. After 10 days of incubation, *Fusarium* cultures developed from each single seed category (symptomatic and asymptomatic) of Titicaca, Rio Bamba, and Real varieties were assigned to a particular “morphotype” according to colony color and shape on PDA, as well as to the morphology of reproductive structures as observed by microscopic analysis (Axiophot, Zeiss). This selection allowed us to obtain a subset of isolates composed of one representative isolate per morphotype. These representative isolates (four in total), after obtaining monosporic culture, were placed into new PDA plates at 22 °C in the dark for two weeks. Their mycelium was then scraped from the PDA and placed into 2 mL sterile plastic tubes (Eppendorf, Hamburg, Germany) at −80 °C, freeze-dried with a lyophilizer (Heto Powder Dry LL3000; Thermo Fisher Scientific, Waltham, MA, USA), and the mycelium was finely ground with a grinding machine (MM60, Retsch, Dusseldorf, Germany) for 5 min with a frequency of 25 Hz. DNA extraction was performed as described in Reference [63], with modifications reported in Reference [37]. Extracted genomic DNA was visualized on a 1% agarose, trizma base-glacial acid acetic-ethylenediamine-tetraacetic acid disodium salt dihydrate (TAE; all from Sigma Aldrich, Merck KGaA, St. Louis, MO, USA) gel in TAE buffer (1X) containing 500 μL/L of RedSafe (iNtRON Biotechnology, Burlington, MA, USA). DNA fragments were separated in 10 cm-long agarose gels, with an electrophoresis apparatus (Eppendorf), applying a tension of 110 V for ~30 min. Electrophoretic runs were visualized using an ultraviolet transilluminator (Euroclone, Milan, Italy). DNA concentration was estimated by comparison with a 1 kb gene ruler (Thermo Fisher Scientific, Milan, Italy). DNA was diluted in DNase-free sterile water for molecular biology use (5prime, Hilden, Germany) to obtain a concentration of ~30 ng/μL and stored at −20 °C until use.

The DNA extracted from *Fusarium* representative isolates was subject to partial *tef1α* gene amplification, purification, and sequencing. A PCR protocol was adopted using a total reaction volume of 50 μL. Each reaction contained 29 μL of sterile water for molecular biology use, 5 μL of 10X Dream Taq Buffer + magnesium chloride (Thermo Fisher Scientific), 3.75 μL of cresol red (Sigma Aldrich), 5 μL of dNTP mix 10 mM (Microtech, Naples, Italy), 2.5 μL of 10 μM EF1 and EF2 primers [64,65], 0.25 μL of 5 U/μL Dream Taq Polymerase (Thermo Fisher Scientific), and 2 μL of template DNA. The PCR cycle consisted of an initial denaturation step (94 °C for 5 min), followed by 30 cycles of denaturation (94 °C for 1 min), annealing (53 °C for 1 min) and extension (72 °C for 1 min), and a final extension (72 °C for 10 min). PCR assays were performed on a T-100 thermal cycler (Bio-Rad, Hercules, CA, USA). PCR fragments were visualized on TAE 1X agarose gel (2%) containing 500 μL/L of RedSafe. DNA fragments were separated at 110 V for ~40 min. Electrophoretic runs were observed with an ultraviolet transilluminator. The size of the amplified fragments was obtained by comparison with HyperLadder 100–1000 bp (Bioline Meridian Bioscience, Cincinnati, OH, USA). PCR fragments were purified and sequenced by an external sequencing service (Genewiz Genomics Europe, Takeley, UK). The sequences obtained were verified by Chromatogram Explorer Lite v4.0.0 (Heracle Biosoft srl 2011) and analyzed by the BLAST database (National Center for Biotechnology Information, http://blast.ncbi.nih.gov (accessed on 18 March 2021)). The sequence of the most representative *Fusarium* isolate (denominated FIESC-PG-Q1) was used to build a phylogenetic tree. Phylogenetic analyses were performed by MEGA software version 7.0 [28] according to [25] by using partial *tef1α* region sequence of the FIESC-PG-Q1 obtained in the present study and of validated phylogenetic species of the FIESC reported in GenBank (Appendix A); *Fusarium concolor* Reinking NRRL 13459 was chosen as the outgroup [25]. Sequences were aligned, nucleotide gaps and missing data were deleted, and a phylogenetic tree was built using the neighbor-joining method [66] with the bootstrap test for 1000 replicates [26]. The evolutionary distances were computed using the maximum composite likelihood method [27].

### 4.5. Molecular Identification of the Pathogen Associated with Quinoa Leaves

A combination of visual, stereomicroscopic, and microscopic observations allowed us to hypothesize that the symptoms associated with Regalona variety leaves were caused by a pathogen belonging to the oomycetes class. For this reason, a sample of 10 leaves was randomly collected from sampled Regalona plants, bulked together and placed into a 50 mL sterile plastic tube (Falcon, Corning, Glendale, AR, USA) at −80 °C, freeze-dried with a lyophilizer (Heto Powder Dry LL3000), and finely ground with a grinding machine (MM60, Retsch) for 5 min with a frequency of 25 Hz. A sub-sample of 1 g was placed into a new 50 mL tube (Falcon, Corning) and subject to DNA extraction using the method previously used by Reference [67] for wheat grains and successfully adapted to quinoa leaves. Total extracted genomic DNA was visualized and quantified as previously described (Section 4.4). The whole DNA extracted was subject to partial ribosomal DNA (rDNA) *ITS* gene amplification of oomycetes. A PCR protocol was adopted as previously described (see Section 4.4) using ITS6 and ITS7 primers [68,69] targeting the *ITS* region of rDNA in oomycetes. The PCR cycle consisted of an initial denaturation step (94 °C for 5 min), followed by 35 cycles of denaturation (94 °C for 30 s), annealing (57 °C for 30 s) and extension (72 °C for 1 min), and a final extension (72 °C for 10 min). PCR assays were performed as previously described (Section 2.4). PCR fragments were purified and sequenced by an external sequencing service (Genewiz Genomics Europe, Takeley, UK). The sequences obtained were verified by Chromatogram Explorer Lite v4.0.0 and analyzed by the BLAST database (National Center for Biotechnology Information, http://blast.ncbi.nih.gov (accessed on 19 March 2021).

According to References [49,50], phylogenetic analyses were performed as previously described (Section 4.4) by using partial *ITS* region sequences of the isolate obtained in the present research and denominated PV-PG-Q1 and those of validated *Peronospora* species from *C. quinoa* and other plant species belonging to the genera *Chenopodium*, *Spinacia,* and *Atriplex* in the *Amaranthaceae* family reported in GenBank (Appendix A). The *Peronospora manshurica* (Naumov) Syd. isolate KUS-F17669 from *Glycine soja* Hort. was included as an outgroup [49,50].

Finally, to detect if the pathogen was seed-transmitted, a sample of 10 g of Regalona seeds was randomly collected and finely ground with a grinding machine (MM60, Retsch) for 5 min with a frequency of 25 Hz. A sub-sample of 4 g was placed into a 50 mL tube (Falcon, Corning) and subject to DNA extraction using the CTAB method previously used by [66] for wheat grains and successfully adapted to quinoa seeds. Total extracted genomic DNA was visualized and quantified as previously described (Section 4.4). The whole DNA extracted was used to carry out a PCR protocol as previously described (Section 4.4) but using PV6F and PV6R species-specific primers [22] for *P. variabilis* detection. The PCR cycle consisted of an initial denaturation step (94 °C for 2 min), followed by 10 cycles of denaturation (95 °C for 30 s), annealing (66 °C for 45 s, −1 °C per cycle) and extension (72 °C for 1.5 min), 22 cycles of denaturation (95 °C for 30 s), annealing (63 °C for 45 s) and extension (72 °C for 90 s), and final extension (72 °C for 5 min). PCR assays were performed and amplified fragments were visualized as described before (Section 4.4). DNA from Titicaca seeds and asymptomatic and symptomatic (infected by *P. variabilis*) leaves of Regalona were also added as a control to this PCR assay.

### 4.6. Statistical Analysis

Incidence (%) of asymptomatic and symptomatic seeds is indicated for each variety as the average (±standard error, SE) of three biological replicates. Data were subject to one-way analysis of variance by considering “variety” as a factor and “incidence” as a variable.The germination rate (%) was indicated for each variety as the average (± SE) of three biological replicates (Petri dishes), both for the asymptomatic and symptomatic selected seeds. Data were subject to one-way analysis of variance by considering, within a variety or between varieties, “seed category (asymptomatic, symptomatic)” as a factor and “germination rate” as a variable.The incidences (%) of each fungal genus recovered during the entire survey are expressed as the average (± SE) of three biological replicates (Petri dishes), both for the asymptomatic and symptomatic selected seeds. Data were subject to one-way analysis of variance by considering: within the variety–seed category combination, “fungal genera” as a factor and “incidence” as a variable; within the variety–fungal genera combination, “seed category (asymptomatic, symptomatic)” as a factor and “incidence” as a variable; and within the fungal genera–seed category combination, “variety” as a factor and “incidence” as a variable.The incidences (%) of each *Fusarium* species recovered during the entire survey were calculated as the incidence of isolates belonging to the morphotype from which the identified isolate was sub-sampled and are expressed as the average of three biological replicates (Petri dishes), both for the asymptomatic and symptomatic selected seeds. Data were subject to one-way analysis of variance by considering, within the variety–seed combination category, “*Fusarium* species” as a factor and “incidence” as a variable;The incidence (%) of FIESC recovered during the entire survey is expressed as the average (± SE) of three biological replicates (Petri dishes) both for the asymptomatic and symptomatic selected seeds. Data were subject to one-way analysis of variance by considering: within the variety–seed combination category, “fungal genera” as a factor and “incidence” as a variable; within the variety–fungal genera combination, “seed category (asymptomatic, symptomatic)” as a factor and “incidence” as a variable; within the fungal genera–seed combination category, “variety” as a factor and “incidence” as a variable.

In all cases, to assess pairwise contrasts, Tukey’s honestly significant difference (HSD) (*p* ≤ 0.05) was used. All statistical analyses were performed with the Microsoft Excel Macro “DSAASTAT” ver. 1.0192 [70]. Finally, the correlations between seed germination rate (%) and FIESC incidence (%) in each quinoa variety and seed category were studied using the coefficient of determination (R^2^), followed by a Student *t*-test.

## Figures and Tables

**Figure 1 plants-10-01933-f001:**
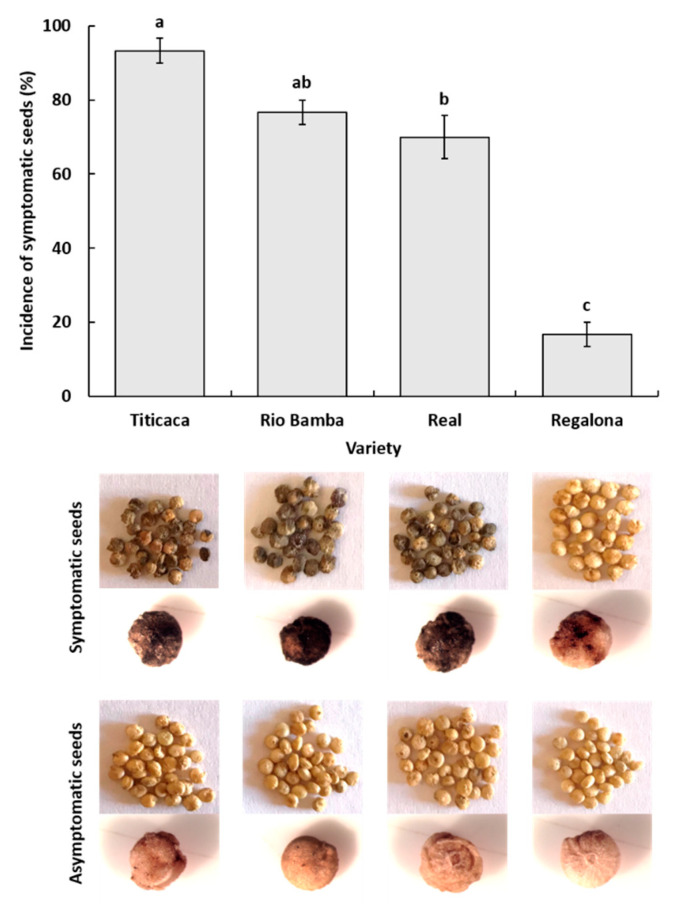
Incidence (%) of symptomatic seeds for each quinoa variety, as assessed by visual and stereomicroscopic observations. Columns represent the average (±standard error) of the three replicates, of 10 seeds each, for a total of 30 observed seeds for each quinoa variety. Columns with the same letter were not different at *p* ≤ 0.05 based on the Tukey HSD for multiple comparisons. Under each column, pictures of symptomatic (browned/necrotic) and asymptomatic seeds of the corresponding variety are shown.

**Figure 2 plants-10-01933-f002:**
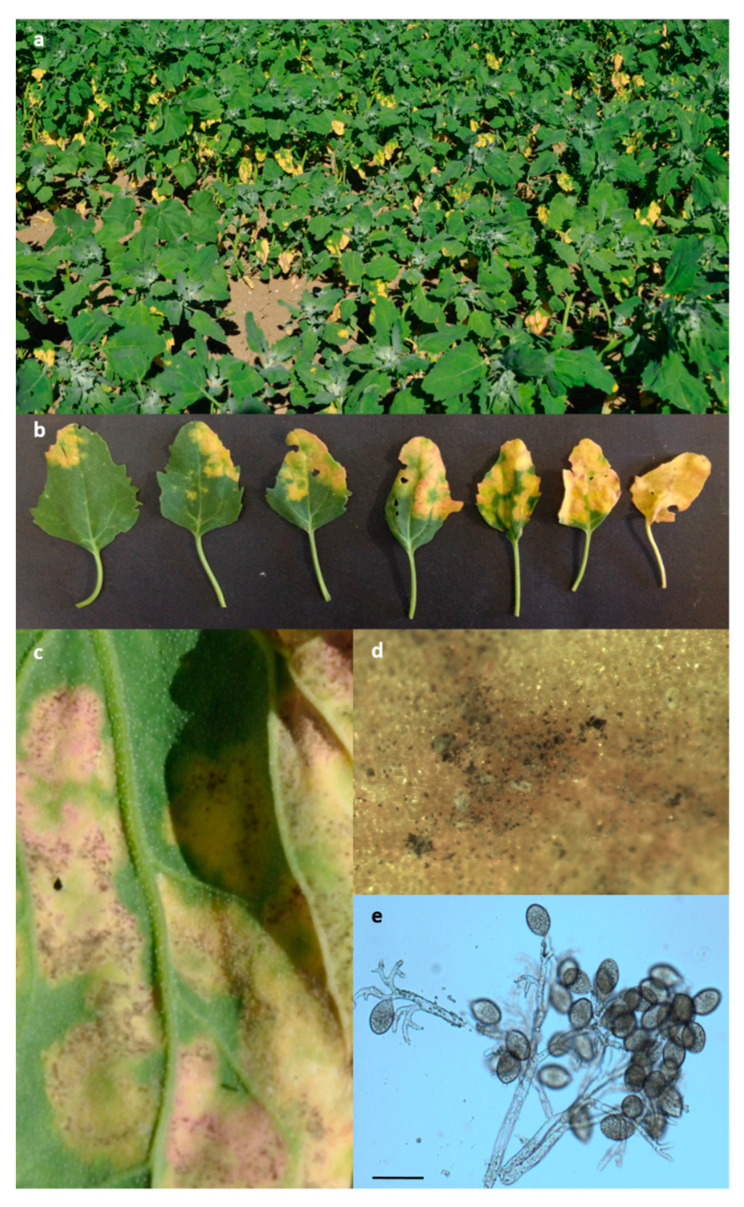
Field view of quinoa plants with foliar chlorosis/yellowing detected in June 2017 on Regalona plots in a trial located at Papiano (Perugia, Umbria, Central Italy) (**a**); yellow chlorotic spots on the upper side of Regalona leaves (**b**) to which corresponded a black-grey efflorescence on the adaxial surface (**c**,**d**) consisting of zoosporangiophores and zoosporangia of the pathogen (**e**, scale bar 50 μm).

**Figure 3 plants-10-01933-f003:**
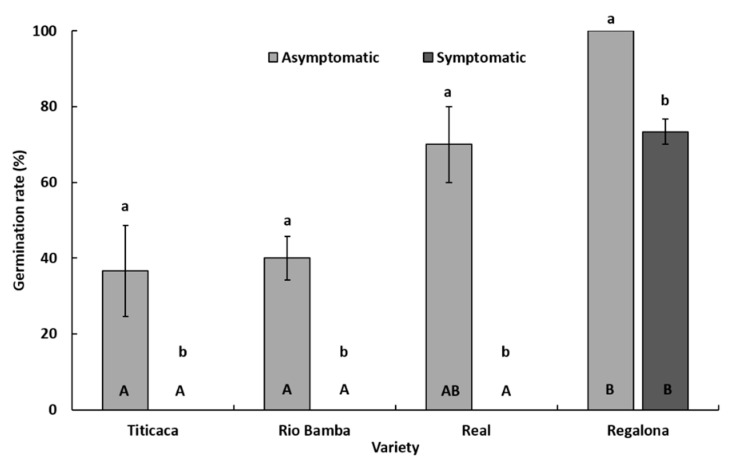
Germination rate (%) of asymptomatic and symptomatic seeds of Titicaca, Rio Bamba, Real, and Regalona varieties. Columns represent the average (±standard error) of the three replicates, each consisting of 10 observed seeds, for a total of 30 observed seeds of each seed category (asymptomatic, symptomatic) per quinoa variety. Within each variety (a and b) or seed category (A and B), averages with the same letter were not different at *p* ≤ 0.05 based on the Tukey HSD for multiple comparisons.

**Figure 4 plants-10-01933-f004:**
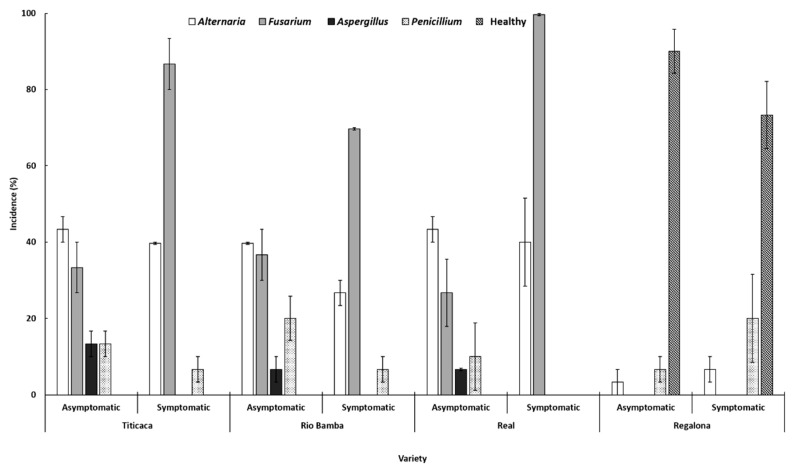
Incidence (%) of different fungal genera in asymptomatic and symptomatic seeds of the four quinoa varieties. Columns represent the average (±standard error) of the three replicates, 10 observed seeds per replicate, for a total of 30 observed seeds of each seed category (asymptomatic, symptomatic) per variety. Data were subject to one-way analysis of variance within the “variety–seed category”, “variety–fungal genera” and “fungal genera–seed category” combinations. To assess pairwise contrasts, Tukey HSD (*p* ≤ 0.05) was used. The main significant differences are discussed in the text.

**Figure 5 plants-10-01933-f005:**
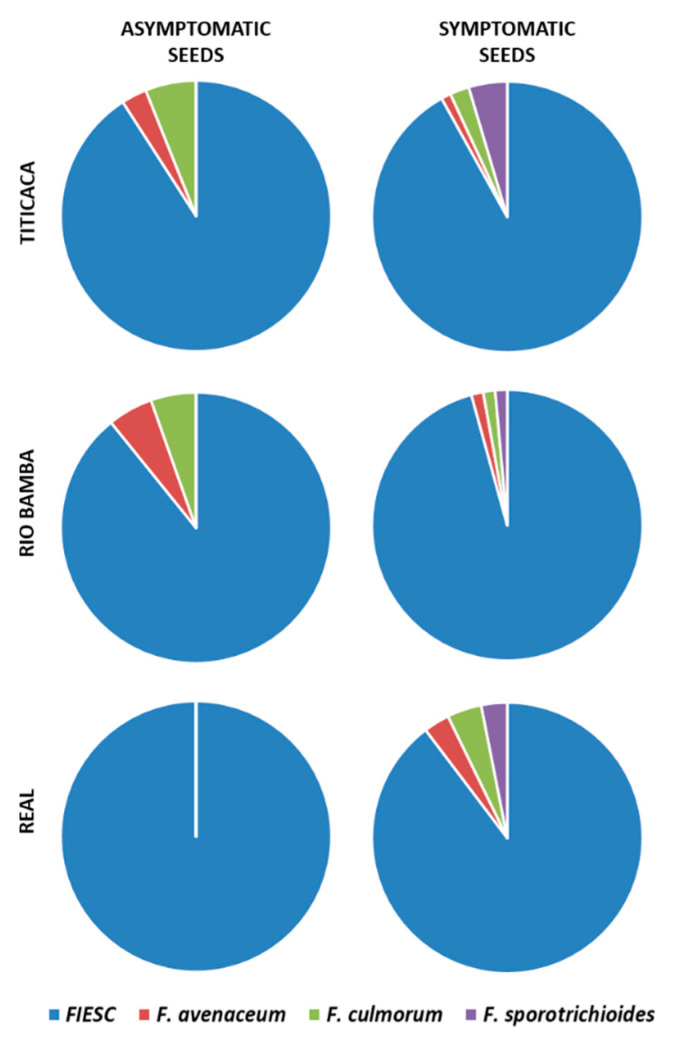
Incidence (%) of the different *Fusarium* species obtained from the asymptomatic and symptomatic seeds of the Titicaca, Rio Bamba, and Real varieties. No *Fusarium* isolates were obtained from the Regalona variety. Pie charts show the average of the *Fusarium* species incidence on the total of *Fusarium* isolates developed from three replicates, with 10 seeds for each replicate, for a total of 30 seeds per seed category (asymptomatic, symptomatic) for each quinoa variety. FIESC = *Fusarium incarnatum-equiseti* species complex.

**Figure 6 plants-10-01933-f006:**
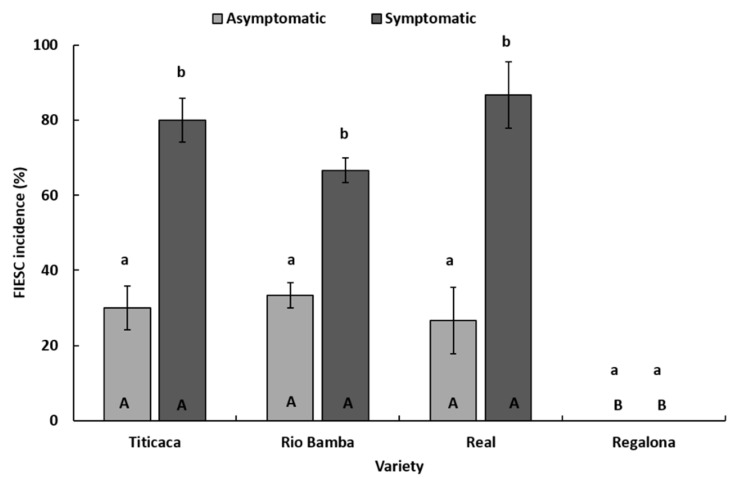
Incidence (%) of *Fusarium incarnatum-equiseti* species complex in asymptomatic and symptomatic seeds of Titicaca, Rio Bamba, Real, and Regalona varieties. Columns represent the average (±standard error) of the three replicates, 10 observed seeds for each replicate, for a total of 30 observed seeds per seed category (asymptomatic, symptomatic) for each quinoa variety. Within each variety (a and b) or seed category (A and B), averages with the same letter were not different at *p* ≤ 0.05 based on the Tukey HSD for multiple comparisons.

**Figure 7 plants-10-01933-f007:**
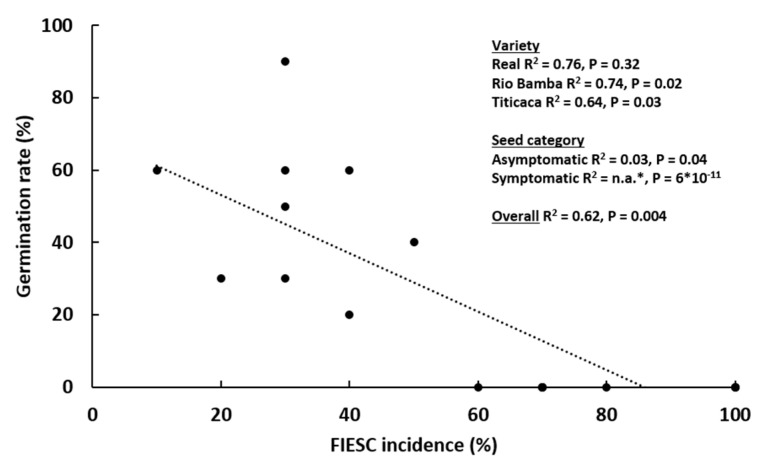
Correlation between seed germination rate (%) of the Titicaca, Rio Bamba, and Real quinoa varieties and *Fusarium incarnatum-equiseti* species complex (FIESC) incidence (%). The three replicates for each “seed category–variety” combination are shown separately. * No germination was detected in symptomatic seeds and no R^2^ was available.

**Figure 8 plants-10-01933-f008:**
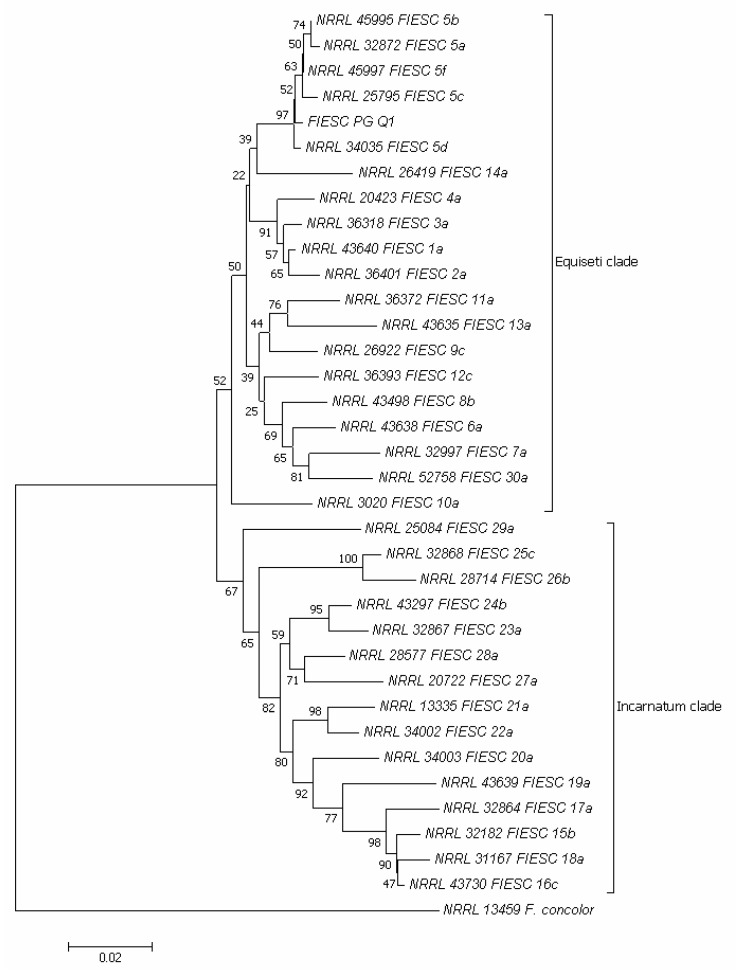
Neighbor-joining tree based on partial *translation elongation factor 1 α* (*tef1α*) gene sequences, showing the phylogenetic relationships between the FIESC-PG-Q1 isolate obtained in the present research and validated phylogenetic species of the *Fusarium incarnatum-equiseti* species complex (FIESC). The *Fusarium*
*concolor* isolate NRRL 13459 was included as an outgroup. The optimal tree with the sum of branch length = 0.64901758 is shown. The percentage of replicate trees in which the associated taxa clustered together in the bootstrap test (1000 replicates) is shown next to the branches [26]. The tree is drawn to scale, with branch lengths in the same units as those of the evolutionary distances used to infer the phylogenetic tree. The evolutionary distances were computed using the maximum composite likelihood method [27] and are in the units of the number of base substitutions per site. The analysis involved 36 nucleotide sequences. All positions containing gaps and missing data were eliminated. There was a total of 545 positions in the final dataset. Evolutionary analyses were conducted in MEGA7 [28].

**Figure 9 plants-10-01933-f009:**
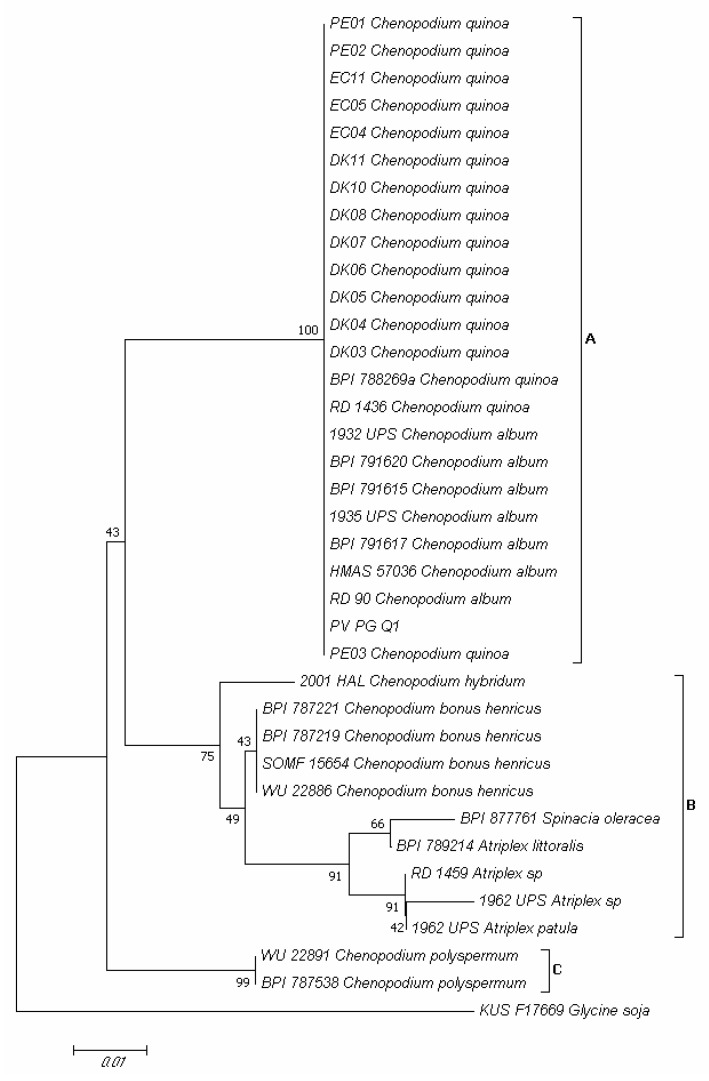
Neighbor-joining tree based on partial *internal transcribed spacer* (*ITS*) region sequences, showing the phylogenetic relationships between the PV-PG-Q1 isolate obtained in the present research from *Chenopodium quinoa* plants of the Regalona variety and validated phylogenetic *Peronospora* species from *C. quinoa* and other plants species belonging to the genus *Chenopodium*. The *Peronospora manshurica* isolate KUS-F17669 from *Glycine soja* was included as an outgroup. The optimal tree with the sum of branch length = 0.19783452 is shown. The percentage of replicate trees in which the associated taxa clustered together in the bootstrap test (1000 replicates) is shown next to the branches [26]. The tree is drawn to scale, with branch lengths in the same units as those of the evolutionary distances used to infer the phylogenetic tree. The evolutionary distances were computed using the maximum composite likelihood method [27] and are in the units of the number of base substitutions per site. The analysis involved 37 nucleotide sequences. All positions containing gaps and missing data were eliminated. There was a total of 229 positions in the final dataset. Evolutionary analyses were conducted in MEGA7 [28].

**Figure 10 plants-10-01933-f010:**
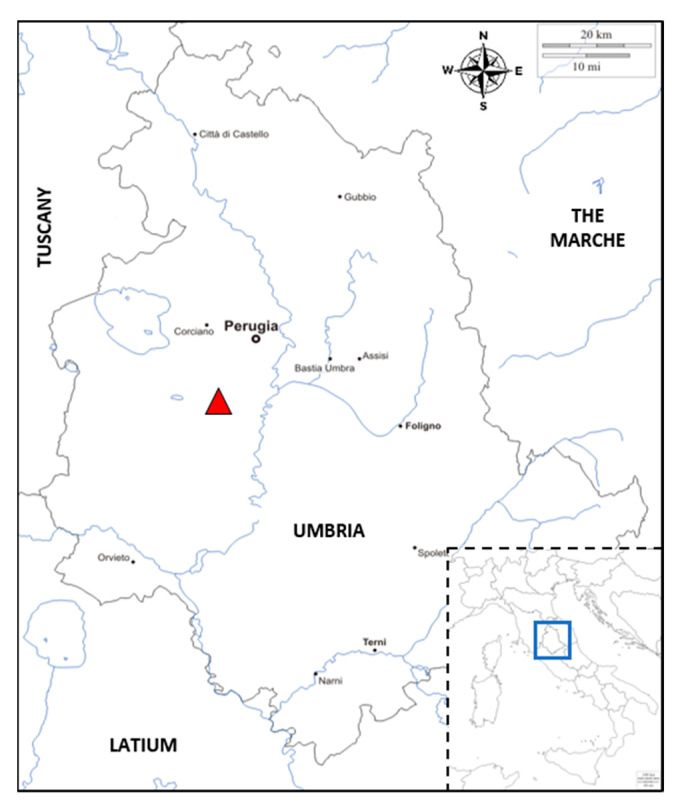
Map of the Umbria region (Central Italy, grey line) showing the experimental field trial location (red triangle) and neighboring regions. In the bottom right corner (outlined box), a map of Italy indicates the region’s location (blue box) in the national geographical context.

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
