# Peer review of "Phytopathological Threats Associated with Quinoa (Chenopodium quinoa Willd.) Cultivation and Seed Production in an Area of Central Italy"

_plants, 2021, doi:10.3390/plants10091933_

Round 1
Reviewer 1 Report
The paper reports investigations on newly observed diseases in plot grown with quinoa in Italy. The introduction clearly explains the importance of quinoa cultivation and the need to monitor the unavoidable emergence of new diseases.
The authors analyzed by diverse methods symptoms in four varieties, either on seeds or leaves. Identification of downy mildew is quite sure, while other symptoms are assigned to a complex of Fusarium species.
The discussion could be more balanced between the two main pathogen groups. A lot of details are given on the Fusarium complex but downy mildew, which appears to be a problem everywhere in the world, could have deserved more attention from the authors.
My main concern about this paper is the analysis of the results of germination and mycological trials. I cannot figure out the rationale of using the "whole" combination as a factor. All seeds of the "whole" group are either symptomatic or asymptomatic, and the general proportion of symptomatic and asymptomatic is presented before. For instance, some 90% of the Titicaca seeds are symptomatic (Fig. 1). No symptomatic seed can germ (Fig. 3). Accordingly, the low germination rate in the "whole" group is caused by the most probable overrepresentation of symptomatic, and therefore non-germinable seeds. Although the samples are independent, their properties (asymptomatic / symptomatic) are not. The differences in the "whole" sample appears to me mostly cause by the unbalanced composition of the sample regarding symptom status.
Unless I completely miss the rationale of the design, I think that the comparison should be restricted to asymptomatic vs. asymptomatic. I believe it won't change the conclusions, or even make them simpler.
Author Response
REQUESTS OF REVIEWER 1 AND POINT BY POINT RESPONSE OF AUTHORS
- The discussion could be more balanced between the two main pathogen groups. A lot of details are given on the Fusarium complex but downy mildew, which appears to be a problem everywhere in the world, could have deserved more attention from the authors.
RESPONSE: we agree with the reviewer, the discussion was unbalanced in Fusarium direction. For this reason, we expanded the considerations relative to downy mildew, in particular those relative to oospores distribution on/in the quinoa seed. We added also 4 additional references and for this reason, the references numbering has been updated in the new version of the manuscript. Please see the additional considerations relative to downy mildew in the discussion section at rows 428-433, 437-452, 468-470 of the new version of the manuscript. - My main concern about this paper is the analysis of the results of germination and mycological trials. I cannot figure out the rationale of using the "whole" combination as a factor. All seeds of the "whole" group are either symptomatic or asymptomatic, and the general proportion of symptomatic and asymptomatic is presented before. For instance, some 90% of the Titicaca seeds are symptomatic (Fig. 1). No symptomatic seed can germ (Fig. 3). Accordingly, the low germination rate in the "whole" group is caused by the most probable overrepresentation of symptomatic, and therefore non-germinable seeds. Although the samples are independent, their properties (asymptomatic / symptomatic) are not. The differences in the "whole" sample appears to me mostly cause by the unbalanced composition of the sample regarding symptom status. Unless I completely miss the rationale of the design, I think that the comparison should be restricted to asymptomatic vs. asymptomatic. I believe it won't change the conclusions, or even make them simpler.
RESPONSE: we addressed the reviewer's indication to reduce the comparison to asymptomatic vs. symptomatic categories. The reviewer is right, this makes the conclusions simpler and doesn’t change the conclusions. We want to bring to the attention of the reviewer that we decided to insert the analysis of the “whole” category to provide a full view of the condition of seed lots without any selection of asymptomatic and symptomatic seeds. However, we recognize that the reviewer comments are all relevant and, for this reason, we remove the “whole” category from the entire manuscript. Of consequence, we modified figure 3, 4, 5, 6, and 7, performed a new statistical analysis (where the “whole” category was present), and delete from the new version of the manuscript all references to the “whole” category (mainly in results and material & methods sections). Please see track changes in the new version of the manuscript.
We are grateful to reviewer 1 for the valuable revision that certainly improves the quality of the paper.

Reviewer 2 Report
This manuscript is well-written and organized.
My only suggestion is that please add how the statistical analysis was performed in the figure legend section of Figure4.
Author Response
REQUESTS OF REVIEWER 2 AND POINT BY POINT RESPONSE OF AUTHORS
- My only suggestion is that please add how the statistical analysis was performed in the figure legend section of Figure 4.
RESPONSE: following the reviewer indication we added in the Figure 4 legend a clarification relative to the statistical analysis performed. Please see the modified version of the legend in rows 200-203 of the new version of the manuscript. To be clear we bring to the reviewer's attention that we did not show the letters because due to the high number of comparisons they would make the figure difficult to understand.We are grateful to reviewer 2 for the valuable revision that certainly improves the quality of the paper.

Round 2
Reviewer 1 Report
The authors have swiftly and dramatically improved the quality of the paper, which appears to me now as fully suitable for publication. I am glad that my suggestions have been useful. The two issues I had raised have been addressed: the questionable (and maybe not required, see below) statistical analysis has been replaced by a more robust one, and the discussion has been balanced.
As for the reply concerning the "whole" category, I understand what the authors attempted to do. This probably would have required a more elaborated statistical analysis, maybe based on probabilistic methods. I don't think, however, this was really worth entering such a complex issue, since the paper still "holds" without it.